# The Emergence of Insect Odorant Receptor-Based Biosensors

**DOI:** 10.3390/bios10030026

**Published:** 2020-03-17

**Authors:** Jonathan D. Bohbot, Sefi Vernick

**Affiliations:** 1Department of Entomology, the Hebrew University of Jerusalem, Rehovot 76100, Israel; 2Institute of Agricultural Engineering, Agricultural Research Organization, Volcani Center, Rishon LeZion 5025001, Israel; sefi@volcani.agri.gov.il

**Keywords:** odorant receptor (OR), volatile organic compound (VOC), biosensor, carbon nanotube field-effect transistor (CNT-FET), indole, skatole, bioelectronic nose

## Abstract

The olfactory receptor neurons of insects and vertebrates are gated by odorant receptor (OR) proteins of which several members have been shown to exhibit remarkable sensitivity and selectivity towards volatile organic compounds of significant importance in the fields of medicine, agriculture and public health. Insect ORs offer intrinsic amplification where a single binding event is transduced into a measurable ionic current. Consequently, insect ORs have great potential as biorecognition elements in many sensor configurations. However, integrating these sensing components onto electronic transducers for the development of biosensors has been marginal due to several drawbacks, including their lipophilic nature, signal transduction mechanism and the limited number of known cognate receptor-ligand pairs. We review the current state of research in this emerging field and highlight the use of a group of indole-sensitive ORs (indolORs) from unexpected sources for the development of biosensors.

## 1. Introduction

In the recent decade, natural volatile organic compounds (VOCs) have been increasingly recognized as valuable diagnostic markers. Detection of VOCs was demonstrated in a wide range of applications with early reports focusing on environmental exogenous VOCs [1] due to their adverse effects on human health [2,3,4]. The possibility of using bacterial VOC biomarkers as a diagnostic approach has drawn much attention lately [5] since different pathogenic species produce a characteristic VOC profile [6,7]. These “bacterial signatures” have been studied using different separation techniques coupled with mass spectrometry [8,9,10].

Analytical chemical tools and particularly mass spectrometry are considered the “gold standard” in this field, largely contributing to the identification of characteristic VOC biomarkers, including specific VOCs of common foodborne pathogens [11]. These methods, however, require cumbersome (and expensive) equipment, labor-intensive preparation steps and trained personnel, and are therefore not suitable for field use. Medical diagnosis has been the primary goal of most VOC detection schemes. Diagnostic devices have been developed for the detection of infectious diseases [12,13], volatile cancer biomarkers [14,15,16] and various metabolic disorders [13]. Analytical assays and diagnostic outputs are required, nowadays, in almost every field of life, necessitating the use of numerous optical and physicochemical methods, applied by different devices. The current trend, however, is geared towards high-throughput, user-friendly and inexpensive small form factor high sensitivity odorant sensors.

Diagnostic devices still heavily rely on optical transducers that apply various methods such as fluorescence, absorbance, surface plasmon resonance, waveguide light spectroscopy, etc. These powerful methods have yielded a wealth of bioanalytic information due to their sensitivity, specificity and relative ease of use. Among all non-optic-based methods that apply rapid and non-invasive on-site analysis, “E-noses” are considered particularly suited due to their sensitivity, low cost of manufacture and ease of use. “E-nose” technologies generally include devices based on conducting polymers, surface acoustic waves, quartz crystal microbalance, metal oxide semiconductor and others [17]. Some graphene-based E-noses have reportedly demonstrated low parts per trillion (ppt) detection limits for nitric oxide (NO) [18], while other field-effect transistor (FET)-based sensors have shown a detection limit in the low parts per billion (ppb) range for xylene [19]. The cost of E-noses dropped, within a decade from thousands of dollars to less than $200 [20], and it is expected to further drop in the near future. There are many examples of successful proofs-of-concept, including devices for an early detection of infectious diseases [21]. The main drawbacks of the “E-nose” are low specificity, inability to distinguish between analytes in a complex mixture, sensitivity to ambient conditions namely moisture and a challenging data analysis. New methods and technologies for the detection of VOCs have made tremendous progress but are still far from exploiting the full potential of micro and nanoelectronics.

An exciting biomimetic strategy, the so-called “bioelectronic nose” offers to combine biological odorant receptors (ORs) with solid-state electronic transducers to produce a hybrid bioelectronic sensor that enables the detection of odorants and the transduction of the chemical signal into an electronic readout. The successful development of small form factor OR-based or -inspired VOC biosensor technology would find uses in a wide range of process manufacturing industries (Figure 1). In an attempt to encompass all of the required disciplines involved in bioelectronic noses, several papers have reviewed lipid-bilayer-based sensors (including nanopores) and cell-based sensors [22], while some have focused on OR-based biosensors [23]. Among the various aspects associated with bioelectronic OR-based sensor development, the integration of solid-state transducers with insect ORs and the transduction of ligand binding into electronic signal are the main focus of this minireview.

## 2. Development of Bioelectronic Sensing

### 2.1. Bioelectronic VOC Sensing

Bioelectronic sensors comprising an electronic (e.g., transistors) or electrochemical transducer functionalized with a bio-recognition element are uniquely suited for modern on-site diagnostic devices. The major advantage is the direct electronic transduction of specific binding into electrons. Direct electronic transduction avoids the use of optics and light sources and allows low-form-factor devices as well as delivers signal levels that are orders of magnitude higher than those achieved with optical sensors. Devices based on electronic platforms are attractive since they are amenable to miniaturization and can be manufactured using conventional microelectronic fabrication techniques. The biorecognition elements available for bioelectronic noses are: (*i*) whole cells (or even tissue such as insect antenna), (*ii*) olfactory receptors or odorant-binding proteins, and (*iii*) synthetic peptides [24].

### 2.2. Cell-Based Bioelectronic Nose

One of the most studied approaches uses cells as the sensing element. In such a case, the original cilia-derived cells are incorporated within the sensor or alternatively, ORs are heterologously expressed in routinely employed expression systems such as yeast, the moth *S. frugiperda* (*sf* cells) or *Xenopus laevis* oocytes. Whole cells containing the desired olfactory receptors were among the first bioelectronic nose examples using crude bullfrog cilia preparation immobilized on a piezoelectric transducer [25]. Other frequently used bionic hybrid systems include human embryonic kidney cells (HEK) expressing the OR of interest. Rat olfactory receptors expressed by HEK cells were integrated with several different transducers such as quartz crystal microbalance (QCM), microelectrodes and surface plasmon resonance [26,27,28]. Human ORs have also been expressed in whole cell-based bioelectronic noses employing HEK cells as vectors [29] or yeast cells attached to gold microelectrodes [30]. The advantages of using cell and even tissue-based bioelectronic sensors are directly related to the complex signal transduction associated with olfaction. By using the OR in its natural environment both the structural connections and coupled cellular downstream elements are preserved [23]. In order to overcome the cumbersome setup of cell-based sensors, portable fluidic devices have been developed [31] and the cell transducer interface has been studied resulting in increased adherence [32]. Recently, a cell-based odorant sensor array has demonstrated prolonged lifespan by using a specific biocompatible membrane anchoring reagent patterned on Polydimethylsiloxane (PDMS) [33]. The feasibility of the detection of multiple odorants was demonstrated by patterning different *sf21* cell lines expressing various insect ORs spatially separated and fluorescently tagged (with a fluorescent calcium indicator). The odorant specific response pattern exhibited high sensitivity in the detection of 1-octen-3-ol, geosmin, bombykal and bombykol. Nevertheless, cell-based bioelectronic platforms suffer from additional obvious limitations pertaining to cell viability and reproducibility in measurements, inherently low signal-to-noise ratio and manufacturability.

### 2.3. OR-Based Bioelectronic Nose

Bioelectronic OR-based sensors generally utilize an OR embedded in natural (nanovesicles, nanosomes [34,35]) or artificial (nanodiscs [36]) membranes (Table 1). Utilizing partially purified or reconstituted ORs as the biorecognition element offers many advantages over whole-cell systems. Such an approach enables the scaling down of devices and a direct interface with micro and nanoelectronics. Furthermore, maintaining the functionality of single proteins or even protein complexes is less challenging than whole cells in which the receptor functionality is dependent on cell integrity. The ideal choice for biorecognition elements is, therefore, olfactory receptors and odorant-binding proteins (OBPs).

While most published studies have utilized mammalian ORs, either purified and reconstituted or heterologously expressed, as described above, only a few examples in the literature describe the use of insect ORs as the sensing elements in an OR-based biosensor (Table 1). As opposed to the G protein-coupled mammalian ORs that are indirectly linked with ion channels, insect ORs act primarily as ionotropic receptors, which is arguably favorable for sensing applications due to the direct linkage between receptor activation and channel opening [37]. For example, *sf21* cells expressing *Drosophila* OR13a were attached to an exposed gate FET sensor [38]. The cells, immobilized on an Al_2_O_3_ extended gate of CMOS integrated devices, were exposed to the odorants bombykol, bombykal and 1-octen-3-ol and the induced ion influx modulated the recorded drain current. In different studies, microelectrodes were used to record signals from olfactory receptor neurons of blowfly in a setup similar to patch clamp electrophysiological measurements [39]. Vertebrates and insect OBPs consist of two distinct families of structurally compact proteins that carry hydrophobic odorant molecules from the gas phase to the respective membrane-bound receptor across an aqueous environment [40]. The use of OBPs as soluble tethered biorecognition elements has also been demonstrated. Mammalian OBPs are presumably tuned to pheromones, which limits their application in bioelectronic sensors [41]. Insect OBPs, which exhibit superior stability, are assumed to have a broad molecular receptive range [42] and genetic modification has been suggested to improve their binding affinities [43]. Further description of OBPs biotechnology applications and specifically biosensing can be found in the literature [44,45].

### 2.4. The Advantages of Insect ORs as Biorecognition Elements

Heavily dependent on chemical cues, insects have evolved a powerful sense of smell exhibiting remarkable sensitivities and the ability to detect a vast number of volatile molecules. Recently, electrophysiological whole-cell measurements have demonstrated the remarkable selectivity and sensitivity of mosquito receptors exhibiting fast responses to the markers indole and skatole in the parts-per-trillion (ppt) range, as shown in Figure 2 [55]. Insect ORs are heteromeric ligand-gated cation channels composed of two transmembrane heptahelical subunits: a highly conserved OR co-receptor (Orco), operating as a non-selective cation channel and ORx, a highly divergent ligand-sensing subunit [56]. Chemo-electrical transduction occurs at the olfactory sensory neurons located in the insect sensillum and involves some other factors, such as OBPs, which are responsible for odorant transfer from air into the aqueous medium and the ORs. Studies of ORs have been fueled, in part, by the need to develop behavior-modifying drugs (e.g., attractants and repellents) and to understand their pharmacology [57,58,59]. Recently, a cryogenic electron microscopy (cryo-EM) structure of Orco homomer at a 3.5 Å resolution was published [60]. The ion conduction pathway generated by the Orco homotetramer exhibits a complex branched structure where a narrow pore leads to a large vestibule followed by a quadrivial architecture of four lateral conduits diverging from the central pathway and allowing the passage of cations (Figure 2a). The distribution of conserved sequences of both Orco and OR along the membrane axis provides insights into the remarkably modular assemblies of Orco with the highly variable OR. More importantly, this structural study emphasizes the vital role of the receptor environment that supports lipid-sequestered, loose transmembrane domains with only a single helix contributed by each subunit to form the central pore. Similarly, the extracellular loops are formed by loose packing of six helices responsible for a large diversity of binding sites. In contrast, densely packed helices form the cytosolic anchor domain. The potential use of insect ORs as biorecognition elements in a sensor faces major challenges, such as: maintaining the conformational behavior involved in the activation of this ion channel, efficiently transducing a binding event into an electronic signal and retaining the binding affinity while the receptor operates outside its natural environment.

## 3. OR-Based Biosensors

### 3.1. Solid-State—OR Interfacing

The integration of solid-state materials and biological systems is still challenging due to their largely dissimilar physical and mechanical properties. Advances in material sciences and microelectronics have enabled fabrication of such integrated devices either by using nanoscale transducers, thus circumventing the “form-factor mismatch” between biomolecules and solid-state or alternatively, by using biocompatible materials. Solid-phase assays such as microarrays have traditionally employed binding proteins or nucleic acids covalently attached to a chemically modified surface and operating in an aqueous environment. Despite altered binding kinetics, surface-immobilized biomolecules retain their residual activities and have been used widely in many diagnostic devices, including such that employ matrix-embedded dried biomolecules. Interfacing membrane proteins with solid-state transducers, on the other hand, presents difficulties that are yet to be fully resolved. The challenging task of protein expression and reconstitution is not trivial as most OR expression systems have a rather low yield. In addition, ORs are membrane proteins and as such, depend on a lipid bilayer environment to maintain structural stability and functionality. Various strategies have been developed to facilitate the incorporation of ORs in sensors while retaining their stability and conformational dynamics.

### 3.2. Nanodiscs

Nanodiscs are self-assembled phospholipid bilayer structures surrounded by membrane scaffold proteins. This technology, recently developed by Silgar and co-workers [61,62], has been proposed for the study of lipid bilayer-based nanodevices or membrane protein reconstitution and is considered an attractive approach for interfacing ORs and micro/nano-electronic measurement platforms. Nanodiscs are generally comprised of artificial discoidal phospholipid bilayers that are similar to high-density lipoproteins encapsulated by one of many available constructs of membrane scaffold proteins (MSP) [63]. A major advantage is the ability to control the size and uniformity of nanodiscs by tuning the lipid-protein molar ratio and by incorporating specific phospholipids and MSP with the appropriate length. The diameter of bare nanodiscs (not containing proteins of interest) can be tuned between 9.5 to 17 nm. Different methods have been used to characterize membrane protein-containing nanodiscs, such as dynamic light scattering, measurement of sedimentation velocity and analytical ultracentrifugation [64]. These characterizations are generally followed by functional studies of the receptor or other membrane proteins. Nanodisc preparation methods including detailed protocols of membrane protein assembly have been published [65]. OR-embedded nanodiscs were previously interfaced with electronic transducers functioning as bioelectronic sensors. The resulting devices have demonstrated enhanced stability and longevity. In particular, different mouse ORs were incorporated in nanodiscs and covalently bound to carbon nanotubes (CNTs) functioning as conducting channels in a FET configuration. The transduction of odorant binding into electronic readout (i.e., conductance changes) has shown reproducible responses over long periods of days and remarkably, even weeks [36]. In another example, researchers fabricated nanodiscs containing the TAAR13c G-protein coupled receptor (GPCR) from zebrafish, which functions as a highly selective OR for cadaverine [46]. The nanodiscs were immobilized on floating electrodes of a CNT-FET and the resulting bioelectronic nose exhibited a dose-dependent response in the form of increased conductance following repeated exposure to cadaverine concentrations as low as 10 pM. The response was measured as a percent change in transconductance compared with the baseline, and although subtle, the changes were reproducible and clearly specific. Application of the device in food spoilage was also demonstrated, as shown in Figure 3. In a recent publication by the same group, a human OR was used for the electronic detection of a rose scent with a reported sensitivity in the fM range [47]. Biosensing with nanodisc-embedded insect ORs was recently demonstrated using multiple *Drosophila melanogaster* (fruit fly) ORs: OR10a, OR22a, OR35a, and OR71a. The nanodiscs were immobilized onto CNT-FET via physical adsorption and challenged with the cognate ligands: methyl salicylate, methyl hexanoate, trans-2-hexen-1-al, and 4-ethylguaiacol. Ligand concentrations of 1 fM were detected [48].

### 3.3. Nanovesicles

Cell membranes containing the OR of interest can be processed into nanoscale vesicles that, depending on their size, maintain planar configuration. Nanovesicles are considered an attractive strategy since they retain the original, natural environment of the receptor. The directed attachment of these small membrane fragments to the transducer largely depends on the membrane’s planar configuration. It is therefore paramount to understand the conditions allowing for a bilayer membrane fragment to maintain planar configuration in solution. The dynamics of lipid bilayer vesicles has been extensively studied using suspended lipid bilayers (SLB) as a model [66]. Vesiculation or planarization of a bilayer membrane is dependent on the free energy of each state, which is in turn, dependent on the entropy of closure and the membrane bending and contour (edge tension) free energies. The bending energy per unit area is given by: $e_{b}=\frac{1}{2}\kappa {\left(c_{1}+c_{2}\right)}^{2}+\bar{\kappa }(c_{1}c_{2})$ where κ is the bending rigidity and $\bar{\kappa }$ is the bending modulus [67], which indicates the membrane malleability. The principal curvatures, *c*_1_+*c*_2_ are the eigenvalues of a curvature tensor that describes the local shape of the membrane. The structural origin of edge tension (γ, the contour energy per unit length) arises from the deformation of the lipids that occupy the edge [68]. It has been suggested that the stability of a planar or a spherical lipid bilayer can be described by [69]: $\alpha =\left(\frac{\gamma }{\kappa_{b}}\right){\left(\frac{A}{\pi }\right)}^{0.5}$, where γ is the edge tension, $\kappa_{b}=2\kappa +\bar{\kappa }$ describes the bending free energy and *A* is the area of the membrane. Using empirical values of $\kappa_{b}$~5-25 *k_b_T* (*k_b_* is Boltzmann constant) and γ=1-2 *k_b_T/l* (*l* is length), it follows that planar lipid bilayers < 50 nm are sufficiently stable. In addition, incubation of the membrane fragments with the transducer above their transition temperature may further facilitate their correct orientation [70]. Generation of uniform and controlled nanovesicles was previously demonstrated using a silicon cell-slicing device comprised of silicon nitride cantilever blades and microfluidic channels. Nanovesicles containing human ORs were produced and characterized prior to immobilization on sensor chips [71]. The reported homogenization method produced nanovesicles of ~100 nm in diameter, as shown in Figure 4, carrying but a few ORs per nanovesicles.

### 3.4. Suspended Lipid Bilayers

Artificial lipid bilayers have long been employed in ion-channel recordings [72] and nanopore applications [22]. Biosensing platforms comprised of hybrid SLB and CNT-FET devices have been considered attractive [73]. Studies of SLB-CNT hybrids [74,75,76] have shown promise in electrical detection of target binding. The shift in the transistor threshold (Δ*V_e_*), due to additional charges, was shown to be related to its charge density by: σ = 2 Δ*V_e_ε_w_ε_0_/λ_d_* where *ε_w_* is the dielectric constant of water and *λ_d_* is the Debye length [77]. The detection of VOC using a nanopore sensor based on SLB has been demonstrated. The reported device utilized the “traditional” cis-trans chamber configuration with an α-hemolysin nanopore embedded in an SLB separating between them. Using agarose in the cis side, which facilitated the dissolution of the volatile pesticide omethoate, and specific aptamers, blocking currents were recorded that corresponded to omethoate concentrations in the ppb range [78]. Insect ORs are attractive candidates for SLB-based nanopore sensors since they function as ligand-gated ion channels with a wide range of ligands [79]. The incorporation of ion channels is generally preferable in biosensing since they allow for low detection limits. Controlling the orientation of OR within the SLB, however, is still considered a challenge [80]. 

### 3.5. Immobilization Methods

Since bioelectronic sensing requires intimate contact between the biorecognition element and the transducer, the immobilization strategy becomes a fundamental part of the biosensor design. Bio-functionalization can generally be categorized into physisorption or chemisorption based methods. The flexibility and ease of bio-functionalization attained by the adsorption of biorecognition elements is advantageous in sensor design. Furthermore, no potentially damaging chemical modifications are needed. Adsorbed molecules, however, are less stable and in certain cases, negatively affect signal transduction. An example is the biofunctionalization of a CNT-FET. Different versions of CNT-FET sensors have attempted to detect biomolecules adsorbed onto pristine and coated CNTs [81]. Transient non-covalent attachment has been pursued using pyrenes or porphyrins exploiting π-π stacking of these molecules with the CNT carbon lattice [82]. Covalent modification imparts a measurable resistance change in the device by converting carbon bonding from *sp*^2^ to *sp*^3^ orientation. The major advantages of this functionalization strategy are: (*i*) locating the interrogated biomolecule in intimate contact with the charge-sensitive region, (*ii*) locating the biomolecule within a Debye sphere around the CNT sidewall (the requisite proximity scales with a Debye screening length [83]) thus enabling electrostatic modulation of OR binding and conformational changes, (*iii*) ensuring optimal orientation of the OR construct with the CNT sidewall, perpendicular to the cation flow in the channel, and (*iv*) enabling a non-transient, stable OR-CNT hybrid for prolonged measurements.

### 3.6. Chemo-Electronic Signal Transduction

The mechanisms by which OR-based bioelectronic sensors convert chemical signals into measurable readout generally rely on piezoelectric, electrochemical or electronic transduction. QCM has been used extensively in the development of bioelectronic noses [23,84]. Different constructs ranging from OBPs to whole cells were coated on the surface of modified QCM and subsequent binding of odorants induced a change in the resonant frequency of the crystal following: ΔF=−FΔm/Art, where ΔF is the change in resonant frequency, *F* the initial crystal frequency, A is the total surface area, *r* is the density crystal, *t* is the thickness of the crystal, and Δm is the change in mass [25]. *C. elegans* ORs (Odr-10) were immobilized onto the surface of gold QCM by aptamer-assisted immobilization and the response to a diacetyl ligand was recorded [85]. In another study, the rat-derived OR17 was expressed on a gold QCM and the response to octanal was recorded [26].

Electrochemical detection using microelectrodes was reported by several studies. Microelectrode array is advantageous since it enables high-throughput measurements. Most studies, however, utilized microelectrode array for whole-cell measurements [39,86,87]. Another strategy, well-known for its superior sensitivity, is electrochemical impedance spectroscopy. In electrochemical impedance spectroscopy, the ORs are generally immobilized onto a working electrode in three-electrode configuration while impedance spectroscopy is measured in the presence of target odorants. Specific odorants (heptanal and octanal) have demonstrated distinct effects on the impedance response of an electrode modified with the rat-derived OR17 compared with an analogous human-derived OR. A model was suggested to interpret a single protein impedance result [88,89]. A single protein was mapped into a network of impedances with N nodes, representing each amino acid and a cut off value of *R_c_* representing the linking distance between each pair. When elemental impedance is attributed to each link the protein topology network becomes an impedance network. The elemental impedance between *i-j*th nodes was given by: $Z_{i,j}=\frac{l_{i,j}}{A_{i,j}}\frac{1}{\rho^{-1}+i\varepsilon_{ij}\varepsilon_{0}\omega }$, where $A_{i,j}=\pi \left(R_{C}^{2}-\frac{l_{i,j}^{2}}{4}\right)$ is the cross-sectional spheres of radius *R_C_* centered on the *i-th* and *j-th* node respectively, *l_i,j_* is the distance between centers, ρ is the resistivity (value of = 10^10^ Ωm), *i* = √−1 is the imaginary unit, ε_0_ is the vacuum permittivity, ω is the circular frequency of the applied voltage. By constructing this network, the authors have compared both receptor-induced responses [90]. 

Functionalization of gold electrodes with liposome-embedded insect ORs (OR10a, OR22a, and OR71a) was also demonstrated in the detection of the target odorants methyl salicylate, methyl hexanoate, and 4-ethylguaiacol, respectively [49]. The electrode impedance response, as reflected by an obtained Nyquist plot, indicated high selectivity and femtomolar sensitivity.

Field-effect transistors have been the most commonly used transducers in bioelectronic noses. In particular, nanoscale FETs incorporating nanotubes as the conducting channel have been studied within the context of bioelectronic sensing. Human ORs (hOR2AG1) were immobilized onto polypyrrole nanotubes and assembled in a FET configuration. The described sensors exhibited high specificity towards amyl butyrate, a common fruit flavor reagent [91]. Graphene is one of the most extensively studied transistor channel materials in bioelectronic sensors owing to unique properties such as high charge mobility, structural stability, and flexibility. Graphene FETs were functionalized with multiple human ORs embedded in membrane fractions allowing for the detection of helional and amyl butyrate with high sensitivity [92]. The normalized sensitivity (Δ*I/I*_0_) was calculated according to the following dependence: $N=\frac{C}{\frac{1}{K}+C}$, where *C* indicates the odorant concentration. Thus, *K* values can be extracted from curve fitting. 

Particularly promising are carbon nanotube field-effect devices due to their extraordinary properties making them excellent candidates for exposed gate biosensors [93,94]. CNTs readily form the conducting channel in FET configuration exhibiting an exceptionally high charge carrier mobility and an extremely stable lattice. Most importantly, the density of charge carriers in these one-dimensional (1D) materials are sensitive to charges in the environment, and therefore the conductance can be modulated by adsorbed molecules. In addition to their biocompatible all-carbon composition, their dimensions are comparable to the size of single biomolecules thus solving the typical problem of “form factor mismatch” between biology and solid-state interfaces [95,96]. Finally, CNT-FET devices are attractive since they are manufactured using traditional microelectronic fabrication techniques [97]. The feasibility of CNT-FET biosensors was demonstrated in the detection of nucleic acids [98,99,100,101] and various protein biomarkers [102] and they are considered the next generation bioelectronic-based biosensors. Detection of VOCs with CNT-FET devices that were bio-functionalized with ORs for specificity is less common [48]. Human ORs were immobilized on a network of single-walled CNT in a FET configuration and the drain current was measured as a function of time. Changes in CNT conductance were observed in response to odorant exposure at various concentrations. The authors suggested that the signal transduction mechanism is related to the electrostatic effect of cysteine residue electrical charge, which changes upon odorant binding [103]. Nanodisc-embedded mouse ORs were immobilized to a CNT-FET and both current response and threshold potential were measured in response to multiple odorants [36]. A different design utilized OBP or alternatively, synthetic peptides conjugated to the FET transducer. One example was the reported use of peptides extracted from the sequence of a *Drosophila* OBP sensitive to alcohols. The peptides were immobilized on a CNT-FET and exhibited a highly selective response towards 3-methyl-1-butanol, the main VOC released from *Salmonella*-contaminated meat [104].

In a single-walled CNT-FET, the ligand binding kinetics are dependent on *D*, diffusion coefficient and *µ*, electrophoretic mobility. *µ* is determined by *V(r)*, the electric field at a distance *r* from the nanotube. For electrolytically-gated CNT-FETs, *V(r)* is determined by the Debye–Huckel model [83] for a point charge *Q* as $V\left(r\right)=\frac{Q}{4\pi \varepsilon_{0}}\frac{1}{r}e^{-r/\lambda_{d}}$, where *ε*_0_ is the permeability of free space and *λ_d_* is the Debye length given by $\lambda_{D}=\frac{1}{\sqrt{4\pi l_{B}\sum_{i}\rho_{i}z_{i}^{2}}}$. *l_B_* is the Bjerrum length, $\rho_{i}$ is the ion density, and *z_i_* is the valence of ion species *i*. *V*(*r*) decreases exponentially with distance over a length scale determined by the Debye length. It follows that lower salt (electrolyte) concentrations will increase the effect of the electric field. Lower water salt content also affects the solubility of a volatile molecule, which is an important consideration in the design of bioelectronic noses, as discussed below.

### 3.7. Phase Transfer in the Electrical Detection of VOCs

Most studies reporting bioelectronic VOC sensing are utilizing a soluble version of the volatile molecule. In nature, the role of transferring VOCs from air to the OR through a mucus layer is accomplished by odorant-binding proteins, which have a high affinity to both ORs and hydrophobic molecules. Incorporation of biorecognition elements within a bioelectronic device inevitably implies an aqueous environment. It should be noted that even in the case of devices that operate in dry conditions there is still a hydration layer that depends on the humidity as well as structural water that is crucial to a protein native conformation. Consequently, several factors need to be considered when designing a device for biosensing volatile molecules. 

The diffusion coefficient of VOCs in air is larger by orders of magnitude compared to water. Mass transfer of a VOC across the air–water interface is dependent on its dissolution rate constant (*k_d_*), surface area of the air-water interface (*A_aw_*), the maximal solubility of the VOC (*C_wmax_*), and the water volume (*V_w_*) such that [105]: $ln\left(\frac{S_{w}-C_{w}\left(t\right)}{S_{w}}\right)=-\frac{A_{aw}k_{d}t}{V_{w}}$, where $S_{w}=C_{wmax}=C_{amax}/H_{c}$, where *C_amax_* is maximal VOC concentration in the air phase and *H_c_* is Henry’s solubility constant. The VOC dissolution rate constant *k_d_* can hence be estimated from the slope of a *ln*(*S_w_-C_w_/S_w_*) versus *t* plot. Hydrophobicity further hinders the partitioning into water of many VOCs. The rather slow VOC dissolution kinetics may pose an obvious limitation on the sensitivity of such sensors. It should be noted that certain types of bioelectronic sensors, such as CNT-FET, have shown the feasibility of detecting single charges and should, therefore, be capable of detecting low concentrations of solubilized VOCs [100]. Issues of poor solubility and high vapor pressure may additionally be alleviated by modifications of the measurement matrix. Various methods of hydrophobic modifications [106] or simply the use of aqueous solutions of organic solvents have been proposed. Such modifications are intended to mimic the role of the OBP (odorant-binding proteins) by increasing the VOC mass transfer as opposed to OBP active transport. 

## 4. Indole-Sensitive ORs (“IndolORs”)

### 4.1. Chemical Description of Indole and Skatole

Indole (IUPAC name 1*H*-indole) with its bicyclic and aromatic structure is the simplest and most ubiquitous representative of its chemical class. It is composed of a six-membered benzene ring fused to a five-membered nitrogen-containing pyrrole ring (Figure 5a). This core scaffold is the basis of a large variety of natural compounds, such as hormones and synthetic molecules of biomedical importance [107]. Due to its predominantly hydrophobic aromatic system, indole (and skatole) is sparingly soluble in water [108]. It exhibits a broad range of biological activities across the animal kingdom and acts as a major interspecies signaling molecule [109]. It is synthesized from tryptophan by bacteria, fungi, yeast, and plants, and depending on its concentration, has a flowery, mothballs, or fecal smell. Indole and its methylated analog, skatole (IUPAC name 3-methyl-1*H*-indole, Figure 5a), are widespread in our everyday life. They are released by wine, meat, dairy products, coffee, seafood, and many other foodstuffs (Figure 5a).

### 4.2. Diagnostic Significance of Indole and Skatole

Detection of indole and its derivative skatole are of paramount importance and their diagnostic value is evident across the agro-food chain, process manufacturing sectors and the clinical arena. 

The organoleptic properties of food depend in part on the presence and respective concentrations of indoles. Our strong attraction to these compounds has promoted their use as additives in personal care products and flavoring agents (e.g., chocolate, coffee, ice-cream, cigarettes and candies, Figure 5b). Indole is a common reagent for the manufacture of perfume, drugs and pesticides. Detecting these compounds is paramount for safety and quality assurance purposes in process manufacturing.

Ensuring food safety and quality is crucial to public health and for limiting food spoilage, respectively. As indicators of decomposition and microbial contamination, indole and skatole have been proposed as quality indicators in marine foodstuff such as crustaceans [110], fish [111], and oyster [112,113,114]. Similarly, indole and skatole may be used as markers of fecal contamination of food [115].

Pork is one of the fastest growing livestock subsectors and demand is rising steeply, particularly in Brazil and China. The presence of the unpleasant odor, the so-called boar taint in pig meat, poses a great risk to the pork supply chain and consumer acceptance is directly dependent on the degree to which it permeates pork meat. Boar taint is perceived as a penetrating “animal-”, “urine-”, “fecal-”, or “sweat-” like unpleasant odor. Indole, skatole, and androstenone are the main contributors to this malodor [116]. Despite the use of chemical analysis tools and sensory panels to detect some of these compounds [117], there are currently no satisfying solutions to detect boar taint in the slaughter line that meet the industry speed, sensitivity, and selectivity requirements [118].

Industrial livestock production is a source of environmental odors, which can be detrimental to animals, workers, and nearby human populations [119]. Skatole has been associated with a variety of health conditions. At high doses, skatole is a pneumotoxin, causing acute pulmonary edema and emphysema in ruminants [120] and damages the lungs and livers of animals and humans [121]. It is also considered a neurotoxic agent, inducing the degeneration of the olfactory epithelium, leading to reversible anosmia in rats [122]. Therefore, skatole may be used as an environmental indicator in farming, in industries relying heavily on this compound, such as perfumery, or in wastewater treatment plants [123].

Human, animal, and agricultural uses of water are increasing globally. Ensuring that water supplies remain safe for consumption and irrigation purposes is a priority that requires the monitoring of indicator organisms such as coliform bacteria, fecal bacteria, salmonella, or *Vibrio cholerae*. These indole-producing organisms can, therefore, be monitored using indole and skatole as biomarkers [124].

Indole and skatole, along with volatile sulfur compounds, are major components of breath odor [125] and halitosis [126] and may be used as clinical markers for the diagnosis of halitosis and underlying etiologies, including metabolic diseases and oral inflammations [127]. Patients with bowel cancer have higher fecal skatole content than healthy individuals [128], suggesting that skatole may be a valuable biomedical marker in this context as well.

In conclusion, the ability to detect indole and skatole with high sensitivity and selectivity offers wide-ranging applications (Figure 1). Such desirable characteristics are only offered by biological systems and currently, the mosquito indolORs are promising candidates for fulfilling these tasks.

### 4.3. Discovery and Pharmacological Properties of IndolORs

The last twenty years have witnessed the discovery of insect receptors beginning with those initially described in *Drosophila melanogaster* [129,130,131] followed by those from several mosquito species [132,133]. Since then, hundreds of insect genomes and several thousand *OR* genes have been annotated representing an unlimited and untapped source of potential biorecognition elements. However, the vast majority of these receptors remain orphan and have no foreseeable use in biosensing. Deorphanized ORs from *D. melanogaster* exhibit various degrees of promiscuity [134] towards high concentrations of compounds of limited interest in biosensing.

The first large-scale deorphanization efforts of mosquito ORs [135,136] led to the identification of the indole receptor named OR2. The closely related paralog OR10 is a skatole receptor in several mosquito species [57,137,138]. OR9, a third member of this gene subgroup [133] was also shown to act as a skatole receptor [55]. Interestingly OR2 and OR9/OR10 exhibit reverse selectivity for the two closely chemical analogs indole and skatole at concentrations that can reach the upper picomolar range based on cell-based assays. These pharmacological features are equivalent or better than those of pheromone receptors [139], which are the epitome of odorant receptor sensitivity and selectivity. The advantage of indolORs, as opposed to pheromone receptors, is their selective and sometime specific relationship to indole and skatole, two compounds of significant interest in many research fields and industries.

### 4.4. IndolOR-Based Biosensor

The mosquito-derived indolergic receptor OR9, the most sensitive odorant receptor discovered so far, may be incorporated with a CNT-FET device to produce an indole- and skatole-specific bioelectronic nose. By utilizing well-established protocols for heterologous expression, OR9 can be expressed in conjunction with the OR co-receptor (Orco), or alternatively, as a single transmembrane receptor (Figure 6). Purification of OR9-containing small membrane fragments and further immobilization to the sidewall of a CNT transistor channel, via covalent modification, would result in an OR9-based CNT-FET bioelectronic nose.

The immobilization of OR9 versus Orco-OR9 should directly affect the signal transduction mechanism and ultimately the final readout. The inclusion of Orco as an integral part of the described CNT-FET device is expected to result in an overall improved performance (Figure 6b). Orco is a dedicated cation channel and should effectively transduce ligand binding. When using a small band-gap semiconducting CNTs, it is expected that the ligand-induced cation current will increase the CNT conductance at positive gate voltage (when electrons are the dominant charge carriers). Cationic charges generally reduce CNT resistance by creating a lower resistance path for electrons. Similarly, a shift in the threshold gate voltage for conduction is expected to be observed during a current-gate voltage sweep. Such a shift reflects a local gating effect due to charges in proximity to the CNT sidewall. Interestingly, by using an OR-functionalized electrochemical impedance spectroscopy (EIS) biosensor, Khadka et al have shown a synergistic enhancement in the obtained signal and two orders of magnitude improvement in detection range due to Orco/ORx combination versus Orx [140]. 

The presence of Orco, however, is not strictly required for bioelectronic sensing [55]. Ligand binding-induced conformational changes of OR9 may be sufficient to affect local electric fields in the vicinity of the CNT sidewall resulting in altered conductance (Figure 6c). Such sensitivity is attributed to the CNT 1D channel transport kinetics. The different indolOR-based CNT-FET configurations are presented in Figure 6.

## 5. Discussion

Modern diagnostics are facing imminent challenges. The increasing population growth rate, socio-demographic changes in developing economies, and accessibility to information are all setting a high bar for clinical, agricultural, and environmental diagnostic throughputs. The increased productivity in many sectors is not accompanied by proper quality assessment. These societal changes call for the development of a new “diagnostic toolbox” that would enable high-throughput and cost-effective diagnostics, available to different end clients from various sectors. There is a huge potential in biochip-based biosensors that converge biological recognition with microelectronics. In particular, bioelectronic assay platforms are promising tools capable of delivering measurable electric signals in response to ultra-low analyte concentrations. Efforts to develop bioelectronic assays aimed at detecting volatiles were fueled by the enormous diagnostic value of volatile markers. Various conducting materials that change their electrical properties following volatiles adsorption have been extensively studied and were applied as transducers in “E-noses” over a decade ago. Despite the relative commercial success of E-noses, they still have fundamental limitations, namely, low specificity and sensitivity to ambient conditions. These limitations have motivated scientists to seek inspiration in nature and consequently led to bio-inspired engineering.

The natural responses to chemical stimuli can range from chemotaxis to complex behavior patterns but the remarkable selectivity is a common characteristic that evolved across all life kingdoms. In animals, this selectivity is governed by specific biomolecular interactions of designated receptors located in sensory neurons. Chemical sensing in animals can, therefore, be described simply as chemo-electronic signal transduction. Similarly, in order to confer ultrahigh specificity on an electronic sensor, a bio-recognition element needs to be incorporated with the transducer. Bioelectronic sensors based on direct transduction of biomolecular binding into electrons have been developed. Olfactory receptors, mostly mammalian, were utilized as biorecognition elements integrated with different transducers such as quartz crystal microbalance, microelectrodes, and field-effect transistors. As opposed to cell-based sensors, the use of OR enables miniaturization allowing for multiplexing and high-throughput. Once integrated with CMOS technology, thousands of bio-functionalized nanoscale transistors can be embedded in one chip [141]. The biofunctionalization of electronic transducers can be quite challenging as ORs are membrane proteins. Different strategies have been proposed but a standardized methodology offering reproducibility and high signal-to-noise ratio as well as maintaining the receptor functionality and stability is yet to be developed.

The use of mammalian ORs, requires the integration of additional molecular factors, since the activation of mammalian G protein-coupled receptors (GPCRs)-type ORs depend on downstream elements. As an alternative for such a complex biofunctionalization scheme, insect ORs offer a simpler transduction mechanism and are thus better suited for bioelectronic sensors. Research in insect OR-based bioelectronic sensors is beginning to emerge as several insect ORs have been shown to be sensitive and specific to environmentally significant VOCs, such as the mosquito octenol receptor [142] or the fly geosmin receptor [143]. It should be noted that only a few insect ORs have been deorphanized, enabling their use as biorecognition elements. The development of sensors incorporating other insect ORs likely involves orthogonal studies, such as electrical physiology aimed at recognizing receptor-ligand pairs. 

Despite the obvious challenges, it should be noted that the tremendous progress in the development of these unique devices is attributed to interdisciplinary bioengineering efforts. These efforts are largely fueled by new nano and microfabrication technologies, better understanding of molecular electronics, improved protocols for biofunctionalization, and incorporation of biomaterials and solid state. Furthermore, recently acquired knowledge of odorant receptors’ biochemistry has significantly contributed to this exciting field. 

Although the results are promising it is clear that this field of research is in its infancy with many challenges ahead. Looking forward, insect OR-based bioelectronic sensors represent a multidisciplinary solution to some of the major bottlenecks and may pave the way towards a whole new class of diagnostic devices.

## Figures and Tables

**Figure 1 biosensors-10-00026-f001:**
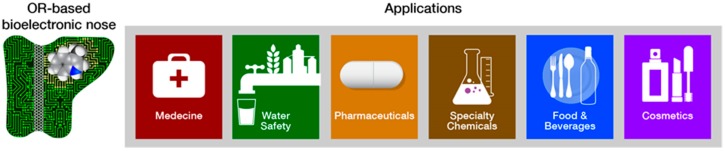
**Odorant receptor** (OR)-based biosensors and their applications. ORs have great potential as biorecognition elements in many sensor configurations. In particular, the integration of ORs and electronic transducers combines the selectivity of the OR with the intrinsic sensitivity of nanoscale solid-state platforms, enabling direct transduction of ligand binding into electronic current. Bioelectronic noses may be applied as accurate, affordable and easy to use diagnostic devices in many fields ranging from agro-food sectors to the clinical arena.

**Figure 2 biosensors-10-00026-f002:**
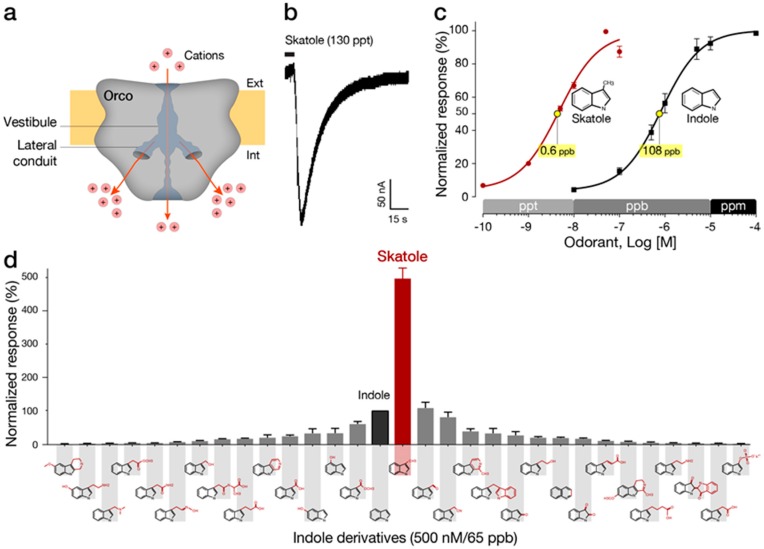
The mosquito OR9 is a supersensitive and specific skatole receptor. (**a**) The ion conduction pathway of the Orco homotetramer (two lateral conduits are shown). (**b**) The novel mosquito skatole receptor (OR9) exhibit fast and robust responses to skatole. (**c**) OR9 detects skatole in the ppt (parts per trillion) range. (**d**) OR9 is highly selective to skatole in the low ppb (parts per billion) range. See Ruel et al., 2019 [55] for additional information.

**Figure 3 biosensors-10-00026-f003:**
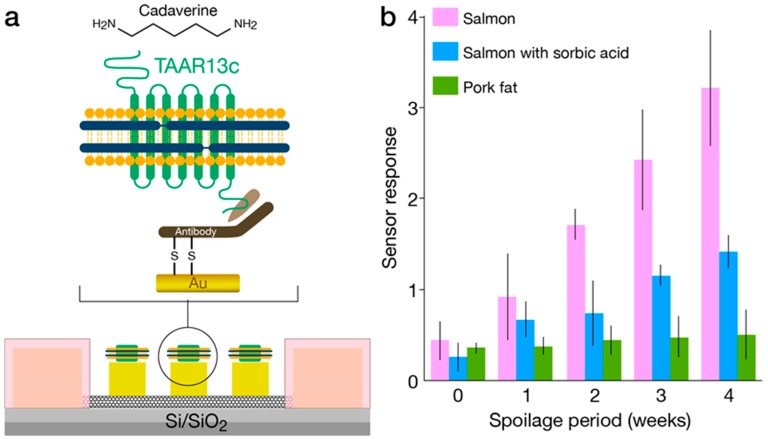
Nanodisc-based bioelectronic nose. (**a**) Diagram of the sensor showing carbon nanotube transistors. The nanodisc-embedded TAAR13c was immobilized onto gold floating electrodes by using a half fragment antibody as a linker. (**b**) The response to the target ligand volatile organic compound (VOC) cadaverine is shown. The percent change in device conductance after exposure to spoiled food samples exhibiting higher responses following prolonged spoilage periods. Adapted with permission from Yang et al. 2017. Copyright (2017) American Chemical Society.

**Figure 4 biosensors-10-00026-f004:**
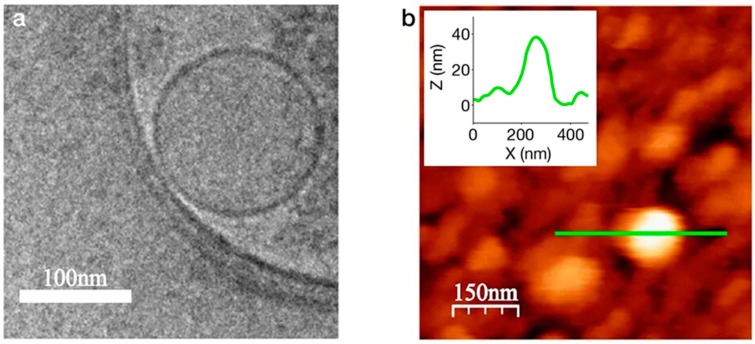
OR-containing nanovesicles. (**a**) Nanovesicles generated from *Saccharomyces cerevisiae* expressing the human OR17-40 are characterized by 2D cryo-EM. Average size of the spheres measures ~ 100 nm. (**b**) An AFM tapping mode image showing the nanovesicles deposited on a gold surface functionalized with –COOH self-assembled monolayer. Reprinted with permission from Sanmartí-Espinal et al. copyright © (2017) [71].

**Figure 5 biosensors-10-00026-f005:**
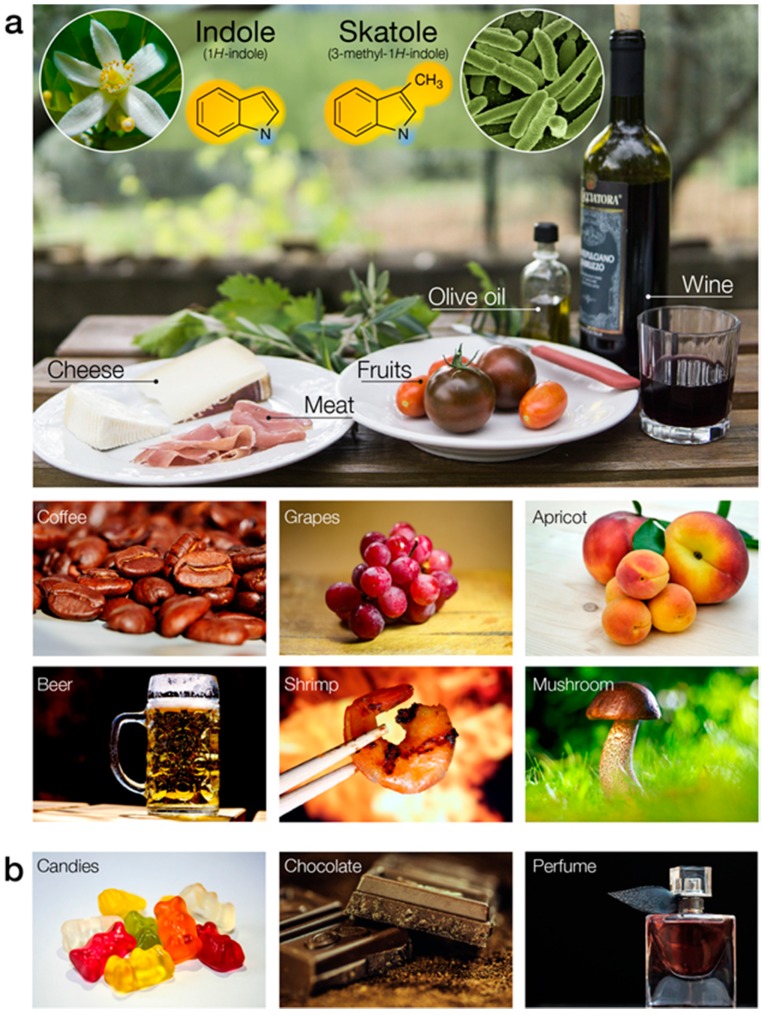
Biogenic origins and occurrence of indole and skatole. (**a**) Indole and skatole are synthesized by the shikimate pathway in plants and by bacterial tryptophan catabolism. These two indolics exhibit a dominant hydrophobic ring system and carbon-hydrogen bonds over a hydrophilic center (N atom). These compounds are naturally occurring in a wide variety of food items (e.g., cheese, meat wine, coffee, grape, apricot, beer, shrimp, mushroom). (**b**) Indole and skatole are used in flavor (e.g., fruity-flavored candies, chocolate) and perfume compositions.

**Figure 6 biosensors-10-00026-f006:**
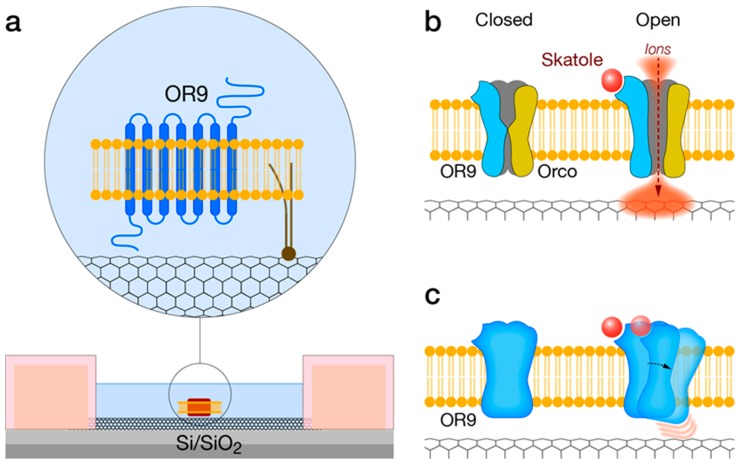
IndolOR-based biosensor. (**a**) Diagram of a bioelectronic sensor showing carbon nanotube field-effect transistor devices. Membrane-enriched OR9 are covalently conjugated to the carbon nanotube sidewall via a phospholipid linker. (**b**) Skatole binding-induced opening of the cation ion channel generates a cation influx transiently locating positive charges near the sidewall, resulting in a significant electric field-effect. (**c**) Conformational changes of OR9 following skatole binding alter charge distribution near the sidewall, which in turn affect the local electric field, resulting in modulated conductance of the CNT.

**Table 1 biosensors-10-00026-t001:** Summary of OR-based biosensors. hOR, human odorant receptor; DmelOR, *Drosophila melanogaster* odorant receptor; MDL, minimum detection limit. Data cited from: Jin et al. [35]; Goldsmith et al. [36]; Yang et al. [46]; Lee et al. [47]; Murugathas et al. [48]; Khadka et al. [49]; Sung et al. [50]; Lim et al. [51]; Oh et al. [52]; Park et al. [53]; and Ahn et al. [54].

OR | VOC		Application	Dynamic Range or MDL	Device	References
ODR-10	Diacetyl	Food & beverages	10^−12^–10^−5^ M	Quartz crystal microbalance	[50]
mOR174-9	Eugenol	Fragrance development	2 ppm	Nanodisc packaged OR-CNT-FETs	[36]
mOR256-17	Cyclohexanone	Process manufacturing	2250 ppm		
hOR 2AG1	Amylbutyrate	Food screening & medical diagnostics	10^−12^–10^−3^ M	Nanovesicle-based OR-CNT-FETs	[35]
hOR 2AG1	Amylbutyrate	Disease diagnostics food safety & environmental monitoring	10^−15^–10^−12^ M	Graphene-based FET	[53]
OR-derived peptide	Trimethylamine	Food screening	10^−15^–10^−4^ M	Nanovesicle-based OR-CNT-FETs	[51]
hOR3A1	Helional	Process manufacturing	10^−7^–10^−3^ M	Liposome-based OR-SPR	[52]
hOR8H2	1-octen-3-ol	Food screening	10^−15^–10^−9^ M	Nanovesicle-based OR-CNT-FETs	[54]
TAAR13c	Cadaverine	food safety	10^−12^–10^−6^ M	Nanodisc packaged OR-CNT-FETs	[46]
hOR1A2	Geraniol	Fragrance development	10^−15^–10^−3^ M	Nanodisc packaged OR-CNT-FETs	[47]
DmelOR10a	Methyl salicylate	Food screening	10^−15^–10^−4^ M	OR/liposome gold sensor	[49]
DmelOR22a	Methyl hexanoate		10^−15^–10^−4^ M		
DmelOR71a	4-Ethylguaiacol		10^−16^–10^−4^ M		
DmelOR10a	Methyl salicylate	Food screening	10^−15^–10^−12^ M	Nanodisc packaged OR-CNT-FETs	[48]
DmelOR22a	Methyl hexanoate				
DmelOR35a	*trans*-2-Hexen1-al				
DmelOR71a	4-Ethylguaiacol				

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
