# Peer review of "The Emergence of Insect Odorant Receptor-Based Biosensors"

_biosensors, 2020, doi:10.3390/bios10030026_

Round 1

Reviewer 1 Report

Authors provide a pretty comprehensive review about insect odorant receptor-based biosensors. They give a clear introduction about insect odorant receptor and their roles in biosensor fabrication in manuscript. The manuscript also focuses challenge using the odorant receptor as bio-sensitive component. Overall, the review will help reader who possibly involved in related work better understand odorant receptor-based biosensors. I would like to recommend its publication in biosensors after a minor revision.

Comment:

In section 1 and 2, authors introduce advantages and disadvantages of different method. It gives a good background about the biosensor investigation. But some detail and exact number are necessary to support some authors’ expressions. Like, ‘“E-noses” are considered particularly suited due to their sensitivity, low cost of manufacture and ease of use.’ What is sensitivity? Detection limit comparison? What is costing difference? In the manuscript, authors give some biosensor examples. Besides introducing transducer and biosensing component in the sensor, can authors give idea which advantage of transducer and biosensing investigator use in the sensor? It will be better if section 4.2 can be organized better. In discussion section, authors declare challenge the sensor are facing. But, what is promising development of sensor? Authors should give more idea about this point.

Reviewer 2 Report

The peer review report for “The Emergence of Insect Odorant Receptor-Based Biosensors”

In this review paper, authors summarize recent research about odorant receptor (OR)-based biosensing technology that enables highly sensitive and selective volatile organic compounds (VOCs) measurement. It is known that the measurement of VOCs is important for a broad area of study related to non-invasive diagnoses such as health status, food safety, water quality, etc. Hence, various kinds of VOC sensing technology have been developed. In particular, the OR-based VOC biosensor is a frontier of VOC biosensor. The authors provide a specific review of insect OR-based biosensors. I guess this review paper will be a suitable one to learn about insect OR-based biosensors. However, I believe that the manuscript can be polished through a revision process. So, I recommend the editor to publish this manuscript after major revision.

Major comments

I feel figure 5 is catchy but not necessary in this scientific review paper. Readers can imagine and understand cheese, meat, fruits, olive oil, wine, coffee beans, grapes, apricot, beer, shrimp, mushroom, candies, chocolate, and perfume without pictures. I recommend to the authors should add another figure to introduce other research paper instead of these pictures. In this review, the authors showed a broad area of applications with OR-based biosensors in Figure 1. However, most of the applications explained in the manuscript are for Food & Beverages. I believe if the authors added explanations of other applications such as medicine, water safety, etc., the significance of this review would increase. I believe that authors should add a table that summarizes measurable VOC, application relevance of VOCs, name, and origin of OR, detection limit, dynamic range, and reference paper of OR-based biosensor. I show an example of a desirable table in Table R1.

Table R1 Summary of a recent study of OR-based biosensor for VOC measurement

VOCs

relevance

Types of OR

Detection limit

Dynamic range

Ref.

ammonia

Kidney disease

ORXX from S. frugiperda

1 ppb

1ppb-100ppb

XX

cadaverine

Food spoilage

TAAR13c from zebrafish

10 pM

10 pM – 1mM

YY

-

-

-

-

-

-

-

-

-

-

-

-

Minor comments

The caption of Figure 6 has an error. I guess it goes outside of the figure box. Please check it. There is some error for the usage of abbreviation and typos. line 140, ORx. line 194, carbon nanotube firstly appeared but did not define abbreviation CNT and is defined at line 200. Also, I found some other errors in the manuscript. Please check through the manuscript carefully again.

Round 2

Reviewer 2 Report

The review report of the revised manuscript entitled "The Emergence of Insect Odorant Receptor-Based Biosensors"

I thank the authors to address each comment and question. I have convinced with the author's comments. I believe that this review has significance in the scope of biosensors.